# *Bertholletia excelsa* Seeds Reduce Anxiety-Like Behavior, Lipids, and Overweight in Mice

**DOI:** 10.3390/molecules26113212

**Published:** 2021-05-27

**Authors:** Oswaldo Frausto-González, Claudia J. Bautista, Fernando Narváez-González, Alberto Hernandez-Leon, Erika Estrada-Camarena, Fausto Rivero-Cruz, María Eva González-Trujano

**Affiliations:** 1Laboratorio de Neurofarmacología de Productos Naturales, Dirección de Investigaciones en Neurociencias, Instituto Nacional de Psiquiatría Ramón de la Fuente Muñiz, 14370 Mexico City, Mexico; ozwafg@gmail.com (O.F.-G.); albertoh-leon@hotmail.com (A.H.-L.); 2Departamento de Biología de la Reproducción, Instituto Nacional de Ciencias Médicas y Nutrición Salvador Zubirán, Vasco de Quiroga 15, Sección XVI, Tlalpan, 14000 Mexico City, Mexico; bautistacarbajal@yahoo.com.mx; 3ISSSTE Hospital Regional “Gral. Ignacio Zaragoza”, Calz. Ignacio Zaragoza 1711, Ejército Constitucionalista, Chinam Pac de Juárez, Iztapalapa, 09220 Mexico City, Mexico; fernandphone_1019@hotmail.com; 4Laboratorio de Neuropsicofarmacología, Dirección de Investigaciones en Neurociencias, Instituto Nacional de Psiquiatría Ramón de la Fuente Muñiz, Calz. México-Xochimilco 101, Col, San Lorenzo Huipulco, Tlalpan, 14370 Mexico City, Mexico; 5Departamento de Farmacia, Facultad de Química, Universidad Nacional Autónoma de México, 04510 Ciudad Universitaria, Mexico; joserc@unam.mx

**Keywords:** anxiety, *Bertholletia excelsa*, central nervous system, lipids, overweight

## Abstract

Overweight, obesity, and psychiatric disorders are serious health problems. To evidence the anxiolytic-like effects and lipid reduction in mice receiving a high-calorie diet and *Bertholletia excelsa* seeds in a nonpolar extract (SBHX, 30 and 300 mg/kg), animals were assessed in open-field, hole-board, and elevated plus-maze tests. SBHX (3 and 10 mg/kg) potentiated the pentobarbital-induced hypnosis. Chronic administration of SBHX for 40 days was given to mice fed with a hypercaloric diet to determine the relationship between water and food intake vs. changes in body weight. Testes, epididymal white adipose tissue (eWAT), and liver were dissected to analyze fat content, triglycerides, cholesterol, and histological effects after administering the hypercaloric diet and SBHX. Fatty acids, such as palmitoleic acid (0.14%), palmitic acid (21.42%), linoleic acid (11.02%), oleic acid (59.97%), and stearic acid (7.44%), were identified as constituents of SBHX, producing significant anxiolytic-like effects and preventing body-weight gain in mice receiving the hypercaloric diet without altering their water or food consumption. There was also a lipid-lowering effect on the testicular tissue and eWAT and a reduction of adipocyte area in eWAT. Our data evidence beneficial properties of *B. excelsa* seeds influencing global health concerns such as obesity and anxiety.

## 1. Introduction

Obesity and overweight are global health issues that are present in many countries, including Mexico, that are in dire need of establishing strategies to control them. Weight loss interventions must take into account comorbidities, such as depression and anxiety, as a possible contributing factor. Several reports suggest a link between anxiety and obesity and overweight [1,2,3], pointing out that anxiety and mental stress increase the risk of developing obesity and being overweight [4,5]. Furthermore, the opposite is also suggested, that obesity is associated with an increased risk of developing anxiety disorder [1], and in overweight women, increased social phobia is reported [6]. Moreover, social anxiety increases the risk of developing metabolic alterations in obese subjects [7]. Treatments that aim to diminish psychiatric pathologies and weight gain and obesity are necessary.

In this sense, one of the strategies is folk medicine; however, studies are needed to support its efficacy and safety. Recently, it was reported that many people use herbal products for weight loss, which include *Bertholletta excelsa* Humb. & Bonpl. (Lecythidaceae) seeds [8].

*Bertholletia excelsa* Humb & Bonpl. (Lecythidaceae) seeds, commonly known as Brazil nuts, belong to an endemic tree of the Amazon Rainforest. Their distribution has been extended to Bolivia, Brazil, Peru, South-Eastern Colombia, Guayana, and Southern Venezuela [9]. Although the tree’s location is restricted to South America, it is acknowledged in different parts of the world due to its different properties with nutritional value and for pharmacological applications [10,11]. It is useful in preparing cosmetics, skin creams, and hair conditioners, as well as an antifungal to control pests in vegetable crops for human consumption [12]. The use of the oil obtained from *B. excelsa* seeds has gained attention because of its antioxidant [13] and anti-inflammatory activities, as well as due to its benefits by supplementing selenium in hemodialysis patients [14]. Its consumption has been related to a reduction in metabolic syndrome characteristics [15]. 

Regarding the pharmacological aspects, its composition of lipids, minerals, and phytochemicals has been associated with improved health parameters; it has shown activity as an anti-inflammatory, antioxidant, and antiproliferative in cancer patients [16] and as a kidney damage mitigator [17]. Moreover, it is considered useful in cardiovascular diseases, such as in hypertension [18], due to the involvement of nitric oxide [19,20], as well as protective in colorectal cancer disease [21,22].

Currently in Mexico and in other countries, Brazilian nut seeds are used as a natural alternative for weight loss in overweight and obese individuals [11]. In Mexico, people who consume this natural product have mentioned that they feel less impulsive to eat, which suggests tranquilizing or anxiolytic-like effects. The effects of this natural resource on the central nervous system (CNS) have not yet been studied. However, this species’ consumption has been associated with restoration of selenium deficiency and a positive effect on the cognitive functions of older adults who suffer mild cognitive impairment [11]. The present work aims to evaluate the neuropharmacological profile of a nonpolar extract of *B. excelsa* seeds to evidence its anxiolytic-like activity and its effects on body weight and lipid metabolism in an experimental model of obesity induced by diet. In addition, the main fatty acid content of the *B. excelsa* seed extract was determined by gas chromatography–mass spectrometry.

## 2. Results

The yield of the *B. excelsa* seed extract (SBHX) was almost 60%. It allowed exploring doses of 3 and 300 mg/kg in the in vivo assays after acute administration. A significant decrease in the behavioral exploration, induced by both SBHX doses, in mice was observed in the open-field (Figure 1A, F_3,16_ = 10.98, *p* = 0.0004) and hole-board (Figure 1B, F_3,16_ = 32.36, *p* < 0.0001) tests, as compared to the vehicle group and resembling the effects produced by the reference drug, diazepam (DZP, 0.1 mg/kg, i.p., Figure 1A,B). Mice were also evaluated in the elevated plus-maze test, in which they spent more time in the open arms, this was significant with the 300 mg/kg dose (Figure 1C, F_3,16_ = 34.29, *p* < 0.0001). 

Administration of sodium pentobarbital (42 mg/kg, i.p.) in the presence of SBHX or DZP (0.1 mg/kg, i.p.) produced a significantly enhanced hypnotic response and lower latency to the sedative effect, whereas low doses of SBHX (3 and 10 mg/kg) resembled the enhanced hypnotic effect of DZP, reinforcing their depressant activity on the CNS (Table 1, F_3,23_ = 8.04, *p* < 0.001). 

As expected, mice that received the vehicle plus the hypercaloric diet showed significant weight gain starting on the first 10 days. In contrast, mice fed the hypercaloric diet but receiving chronic administration of SBHX at 30 mg/kg showed significant weight gain until Day 24, which continued increasing until Day 40 (Figure 2A, left side). Prevention of weight gain was more pronounced in mice receiving 300 mg/kg, showing a significant increase until Day 35 (Figure 2A, right side) (Figure 2A, Treatment: F_2,12_ = 3.711, *p* = 0.05; Time F_2.360, 28.32_ = 28.59, *p* < 0.0001; Interaction: F_16,96_ = 2.38, *p* = 0.005). A significant decrease in weight was observed without alteration in food (Figure 2B) and water (Figure 2C) consumption in both the 30 and 300 mg/kg SBHX groups in comparison to the vehicle group.

After euthanasia, tissues such as liver, testes, and epididymal white adipose tissue (eWAT) were processed to determine the biochemical profile of lipids, as an increase in overweight and obesity was reported. Total testicular lipids showed a significant decrease in mice receiving SBHX (30 or 300 mg/kg) (Figure 3B, F_2,12_ = 9.55, *p* = 0.006). No changes were found in other tissues evaluated (Figure 3A,C–G). In contrast, parameters such as eWAT weight, total lipids, and adipocyte area showed a significant decrease with both doses (Figure 4A, F_2,12_ = 5.50, *p* = 0.01; Figure 4B, F_2,12_ = 20.80, *p* < 0.001; Figure 4E, H:200.49, fd: 2, *p* < 0.001). However, cholesterol (CHOL) and triglycerides (TG) showed a significant decrease or increase, respectively, at a dosage of 300 mg/kg (Figure 4C, t:2.53, *p* = 0.035; Figure 4D, F_2,12_ = 7.22, *p* = 0.009). Figure 4F shows a physical reduction in the amount of adipose tissue after SBHX (30 or 300 mg/kg) administration in comparison to that obtained in mice receiving the vehicle. Adipocyte size was also diminished in a pronounced manner in the group receiving SBHX at 300 mg/kg (Figure 4F, right panel).

Fatty acid methyl esters (FAMEs) were identified by comparing the obtained retention time and mass spectra with those from their standards. The mass spectra were also compared with those of the NIST library. Four major fatty acids were identified in *B. excelsa* oil and the predominant unsaturated fatty acids were oleic (C18:1, 59.97%, T_ret_ 23.537 min) and linoleic (C18:2, 11.02%, T_ret_ 23.485 min) (Figure 5).

The main saturated fatty acids were palmitic (C16:0, 21.42%, T_ret_ 21.864 min) and stearic (C18:0, 7.44%, T_ret_ 23.732 min). Another fatty acid detected in trace amounts was palmitoleic acid (C16:1, 0.14%, T_ret_ 21.684 min) (Figure 6). The composition of the oil was similar to that previously reported by Peña-Muñiz et al. [23]. The chromatogram included in this manuscript shows all the compounds identified and resolved using a methodology reported to identify and estimate the fatty acid mixtures as methyl ester derivatives. The simplest method of FAME quantification by EI-MS can be carried out by monitoring a range of mass-to-charge (*m*/*z*) values that encompass the fragments expected from the analytes and determining the amount based on integrating the peaks into the total ion count chromatogram [24].

## 3. Discussion

It has been reported that overweight and obesity show high comorbidity with anxiety or depression [25,26,27]. In fact, obesity and mental illness are considered as addressing a double epidemic [28]. A review of the PubMed database allowed obtaining at least 2164 reports using the terms “anxiety and overweight” and 3087 using “anxiety and obesity”. However, pharmacological therapy has been focused only on searching options for one or another affection, instead of an integrated treatment or considering mental factors as the etiology of overweight and obesity, or the contrary. In this study, we found, for the first time, to the best of our knowledge, evidence of the health benefit of a nonpolar *B. excelsa* seed extract in producing anxiolytic-like effects, without acute or chronic toxicity, and simultaneously preventing weight gain and reducing obesity signs. Two doses were considered in our experimental design to corroborate not only significant effects but also whether these effects were produced in a dose-dependent manner.

*B. excelsa* seeds are a natural product with biological and nutritional attributes [11,16]. In Mexico, as in other parts of the world, these seeds are consumed in great demand, mainly as a remedy for weight loss [15] associated with calming effects according to anecdotal reports. However, there are not enough preclinical or clinical studies reported in the scientific literature to corroborate these properties. Thus, a neuropharmacological profile of the *B. excelsa* seeds was evaluated in this study to know its beneficial and/or adverse effects produced on the CNS. Additionally, its influence on weight and lipid metabolism was also explored in mice.

Firstly, an oil extract from *B. excelsa* seeds was obtained by using maceration to yield 60% of dry weight. Our data agree with those previously reported as a yield of 60 to 70% for this species [16]. After chemical analysis, the presence of the following fatty acids was confirmed: palmitoleic acid (0.14%), palmitic acid (21.42%), linoleic acid (11.02%), oleic acid (59.97%), and stearic acid (7.44%). This composition also agreed with a preliminary report of the fatty acid composition of the *B. excelsa* oil [23]. Of these fatty acids, linoleic acid is the main one associated with a reducing action of the adipose tissue mass; this has turned it into a nutraceutical (useful food in the prevention or treatment of diseases), as it is considered that it reduces the incorporation of lipids in different tissues, especially adipose [29]. In this sense, inhibition of proliferation and delay in adipose cells (3T3-L1) clustering have been reported as due to the reduction of the levels of PPARγ-2, a nuclear receptor that regulates the formation of adipose or fatty tissue [30,31]. In addition, all these fatty acids found in *B. excelsa* oil, among others, have been detected as a mixture in the human amniotic fluid, colostrum, and milk [32]. This mixture produced anxiolytic-like effects, similar to diazepam, in male and female adult Wistar rats associated with the GABA_A_ receptor [32,33]. The depressant effects were also observed in infant rats by corroborating the GABA_A_ receptor involvement [34].

To assess the depressant activity of the *B. excelsa* oil extract, we explored its effects in different experimental anxiety models. These included the open-field test, the hole-board test, and the well-known and validated plus-maze test. The exploratory nature of rodents allows them to decide to stay and prefer closed spaces to the open and elevated spaces represented in the elevated plus-maze [35], which is recognized as an ideal tool for screening possible anxiolytic-like drugs, where either mice or rats are used. In all models used, *B. excelsa* promoted an anxiolytic-like effect that provided preclinical evidence for its use with clinical implications. It is important to mention that in this preliminary investigation with *B. excelsa*, we used male mice to reduce biological variability due to sex. To our knowledge, females have a close relationship with hormone regulation and CNS alterations but also in their lipid metabolism, food intake, and body weight [36,37]. However, after having obtained a positive anxiolytic-like response, it will be interesting to assess, in future experiments, the effects of this treatment in female mice, as the prevalence rate of anxiety disorders is approximately twice as high in women as in men [38].

To complement the spectrum of the CNS’ depressant activity of SBHX, the hypnotic-like response, induced by sodium pentobarbital in mice, was potentiated by SBHX. As known, substances that act as allosteric agonists of the GABA_A_ receptor can enhance the depressant action of this main inhibitor amino acid in the brain [39]. *B. excelsa* produced a clear hypnotic effect equivalent to that induced by DZP, suggesting a mechanism of action on the GABA_A_ receptor, at least in the regulation of the CNS effects. Specific experiments, in the future, could contribute to corroborate this hypothesis.

As far as we know, this is the first time that the effect of *B. excelsa* on the CNS has been analyzed but also on weight gain and fat accumulation using an animal model of obesity [40]. CHOL and TGs were explored in the total percentage of lipids, because they are two important and major constituents of the lipid fraction of the body [41]. However, it would be interesting to explore other components of the total lipid fraction in future studies, because, according to the present results, we cannot discard that some of them are likely implicated in the effects of *B. excelsa.* Chronic administration of the *B. excelsa* seed extract prevented the body weight gain induced in mice fed with a hypercaloric diet without altering their food or water consumption. Interestingly, a reduction in the amount of adipose tissue and adipose size was quantified too. Total percentage of lipids was significantly reduced in part because of a reduction in TGs; as known, TGs represent the main lipid component of dietary fat and fat depots in animals [41]. The consumption of *B. excelsa* seeds in a group of obese Brazilian adolescent women (*n* = 17) improved their lipid profile and microvasculature function, these effects were associated with the high level of unsaturated fatty acids, among other possible bioactive substances [42]. It has been reported that consumption of Brazil nuts does not affect the biochemical parameters of liver and kidney function, indicating a lack of hepatic and renal toxicity in healthy humans; on the contrary, results indicate a long-term decrease in inflammatory markers after a single intake of a large portion [43]. The excess of macronutrients in adipose tissues stimulates them to release inflammatory mediators, such as tumor necrosis factor α (TNF-α) and interleukin 6 (IL-6), and reduces the production of adiponectin, predisposing to a proinflammatory state and oxidative stress [44]. A single intake of Brazil nuts (20 or 50 g) caused a significant decrease in IL-1, IL-6, TNF-α, and IFN-γ serum levels, whereas serum levels of IL-10 were significantly increased [43]. The consumption of only one *B. excelsa* nut per day for 3 months was effective to reduce the inflammation, oxidative stress markers, and atherogenic risks in patients [14], reinforcing the benefits of the Brazil nut in obesity but also in other metabolic diseases.

On the other hand, unsaturated fatty acids, such as 3-PUFAS and oleic acid, reduce adipose tissue mass [42], particularly in the epididymal tissue. Further, data indicate that oleic acid and 3-PUFAS reduce weight gain through several mechanism, including thermogenesis. In the present study, the extract of *B excelsa* seeds showed a high percentage of oleic (59.97%) and linoleic acid (11.02%, precursor of 3-PUFAS); thus, it is possible that these compounds contribute to the antiobesity effect observed in this study.

Effects of *B. excelsa* on the lipids profile have been indicated already. For example, in a controlled study in patients with hypercholesterolemia, who consumed *B. excelsa* seeds, a reduction in serum CHOL [45] occurred. It is expected that some components of the lipid fraction increase or decrease depending on different factors such as a relationship with the lipid diet content, metabolic aging, and sex [36,37,46]. In our study, an increase in CHOL was observed only in mice receiving chronic administration of the highest dosage (300 mg/kg). CHOL is an unsaturated alcohol of steroid compounds, essential for cell membranes but also a precursor of adrenal and gonadal steroid hormones [41], suggesting its involvement in reproductive activity [46]. Actually, overweight and obesity are related to alterations in the reproductive system [47]. For example, some reports indicate that at the same time that body weight increases, a decrease in testes weight is observed in male rats fed with an obesogenic diet [47,48]. A decrease in sperm quality and viability has been reported with important fertility consequences [48]. The present results showed that the administration of *B. excelsa* prevents an increase in body weight, eWAT accumulation, and lipid content without altering testes’ weight. Therefore, it is possible to suggest that the extract could also help to reduce the fertility problems associated with obesity. The mechanisms of action involved in this effect require further exploration. A limitation of the present study is that the quality of sperm and spermatogenesis was not analyzed. Degeneration of seminiferous tubules has been reported as a consequence of an obesogenic diet (vacuolar changes in seminiferous tubules, spermatogenic cell dysfunction, among others) [48,49], which is reversible by chronic treatment with *Cinnamomum zeylanicum* [50] and metformin [48] and, at the same time, helps to reduce the signs of obesity. After gross analysis, our results showed no changes in the number of seminiferous tubules after chronic SBHX treatment for 40 days.

It has been reported that there is an approximate 25% increase in the likelihood of developing mood disorders and anxiety in overweight and obese people [25,27]. The findings of this study on the anxiolytic-like effect of *B. excelsa,* together with the modulation of overweight involving lipid metabolism, are relevant because of two important reasons. The first is based on the fact that overweight and obesity constitute a syndrome with a high prevalence of psychiatric comorbidity, such as disorders of anxiety, depression, addictive behaviors, and eating disorders, among others [2]. They are considered as one of the main “social stigmas” that trigger psychological disturbances, which increase per se anxiety, affecting the quality of life of the individual.

Secondly, drugs approved for overweight and obesity mainly focus on weight loss and “do not take into account the motivational factors and mental behavior that commonly cause obesity” [51]. In addition to this, many of these medications have the adverse effect of being anxiogenic, for example phentermine/topiramate and naltrexone/bupropion, further increasing the likelihood of presenting this comorbidity and, with it, the drug interaction that complicates treatment.

Therefore, the anxiolytic-like effects of the *B. excelsa* seeds could be useful as an adjuvant in the treatment of overweight and obesity. Nevertheless, since significant enhancement effects on the sodium pentobarbital-induced sedative-hypnotic condition were observed, it is important not to combine therapeutic alternatives without evidence that they are safe.

The median lethal dose (LD_50_) was calculated as a parameter of acute toxicity [52]. It was calculated to be greater than 2000 mg/kg. Thus, the *B. excelsa* seed extract was classified at level 5, suggesting minimum toxicity [53]. It is important to mention that *B. excelsa* seeds are commonly called “Brazilian seed”. Unfortunately, there are other seeds commercialized with the same name that exert toxicological effects, including convulsions or death. In the case of *Thevetia peruviana* seeds [54], their acute oral toxicity was reported to be 447 mg/kg in mice, producing tachypnea, severe hypothermia, muscle relaxation, generalized seizures, and death [55]. Another similar case, important to mention, is the *Aleurites moluccanus* seeds [54]; their acute toxicity is represented by convulsions and death observed at doses of 326 mg/kg, p.o. and 71 mg/kg, i.p. evaluated in mice in our laboratory.

## 4. Materials and Methods

### 4.1. Animals

Male Swiss-Webster (SW) mice (25–30 g body weight, 6 weeks old at the beginning of the study) kept at a controlled temperature of 22 ± 1 °C and light/dark cycles of 12 h were used in groups of at least 5 animals (*n* ≥ 5). The number of animals in each experiment was kept to the minimum necessary to observe significant effects. Exploration was carried out at our institute following the specifications issued by the institutional Ethics and Research Committee with the approval of the Project Number NC-123280.0 and NC-17073.0 (CONBIOETICA-09-CEI-010-20170316) and according to the Official Mexican Norm for the care and handling of animals (NOM-062-ZOO-1999) and the International Guide for the Care and Use of Laboratory Animals (8th Edition, NRC, The National Academy Press, Washington, DC, USA).

### 4.2. Reagents and Drugs

Diazepam (DZP) (Psicopharma^®^, S.A. de C.V., Mexico), sodium pentobarbital (SPB) (Sedalpharma^®^ Agrosur Distribuidora, S.A. de C.V., Mexico), and *B. excelsa* seeds were purchased from SdB (Guadalajara, Mexico) and hexane (High Purity México, S.A. de C.V., Mexico) Drugs were freshly prepared the day of the experiments and dissolved in saline solution. Except for DZP and the hexane extract of *B. excelsa* (SBHX) that was dissolved in 0.2% Tween 80 in saline solution.

### 4.3. Extract Preparation

*Bertholletia excelsa* seed extract was obtained from *B. excelsa* Humb & Bonpl. (Lecythidaceae) seeds sold as a nutritional supplement (SdB, Guadalajara, Mexico, approximately 1.5 g per commercial presentation). Seeds were cut into very small pieces (1610.43 ± 8.53 mg, *n* = 3 batches) to be macerated in hexane (25 mL) for 72 h each time for three simultaneous extractions. After total evaporation of the solvent using rotary evaporation (Büchi R-100, Latinoamérica S. de R.L. de C.V., Mexico) and aeration, a translucent oil was obtained with a 55 ± 2.5% yield. Esterification of the sample was performed by the Khan and Scheinman method, followed by fatty acids determination through gas chromatography. The fatty acid methyl esters (FAMEs) were identified by comparing the retention time and the obtained mass spectra with those from the fatty acid methyl esters (FAME) standard (Figure 5).

#### Fatty Acid Composition of the *Bertholletia Excelsa* Oil

Esterification was performed by the Khan and Scheinmann method, followed by the determination through gas chromatography. Analyses of FAMEs were carried out using a Clarus 660 GC system (Perkin-Elmer, Waltham, MA, USA) equipped with a DB-5 (30 m × 0.25 mm i.d., 250 nm film thickness) capillary column (Supelco, Bellefonte, PA, USA) and a Clarus SQ8C Mass Spectrophotometer, operated in electron-impact ionization mode (70 eV). GC-MS analyses were carried out in split mode (split ratio 1:50), using helium as carrier gas (1 mL/min flow rate). The injector temperature was fixed at 250 °C. The sample volume injected was 1 μL. Oven temperature was held at 40 °C. for 3.5 min and then programmed at 10 °C/min to a final temperature of 300 °C, where it was maintained for 10 min.

### 4.4. Pharmacological Evaluation

#### 4.4.1. *Bertholletia Excelsa* Acute Effects on the Central Nervous System

Experiments were scheduled from 9:00 to 14:00 h. Independent groups of mice received the vehicle, a treatment of SBHX (30 or 300 mg/kg, i.p.) or the reference drug (DZP, 0.1 mg/kg, i.p.). Thirty minutes after administration, the anxiety-like response was analyzed in the following tests using dim red light (30 lux).

##### Open-Field Test

Each mouse was placed in an acrylic box divided into 12 quadrants. The number of squares explored by each mouse was registered for 2 min to declare an anxiolytic-like response in comparison to the reference drug [56].

##### Hole-Board Test

This test consisted of one acrylic box with wooden floor with holes of 3 cm diameter evenly distributed. Each mouse was placed inside the box, and the times the mice introduced the head in a hole were registered. The test was monitored for 3 min [57].

##### Elevated Plus-Maze Test

This test consisted of four wooden arms (two open and two closed) joined in the central part and elevated 50 cm from the floor. Each mouse was placed in the center of the cross. Immediately after, the time of permanence in open or closed arms and the number of crosses in each arm during a period of 5 min was registered [35].

##### Sodium Pentobarbital (SPB)-Induced Hypnosis Potentiation

Mice treated with the vehicle, diazepam (0.1 mg/kg, i.p.), or treatment with SBHX received also SPB (42 mg/kg, i.p.) 30 min later to register the latency to motor coordination (phase of sedation), the latency to the loss of righting reflex (phase of hypnosis), and the duration of hypnosis (time from the start of the righting reflex loss to its recovery) [58].

#### 4.4.2. *Bertholletia. Excelsa* Chronic Effects on the Overweight and Lipids

Independent groups of mice received chronic treatment of either the vehicle or SBHX (30 or 300 mg/kg, i.p.) daily for 40 days. Simultaneously, mice were fed with standard diet Purina 5001 (LabDiet Rodent); its calories are provided by protein (28.67%), fat (ether extract, 13.38%), and carbohydrates (57.94%). Water containing sucrose at 34% *m*/*v* was included to generate a hypercaloric diet, a combination that induces overweight and obesity in mice [40]. The measurement of the body weight of the mice and the intake of water and food was recorded individually per day, the averages were plotted every 5 days for 40 days, taking as zero or baseline the day at which the anxiolytic effect was evaluated.

Several reports indicate that overweight and obesity produce an accumulation of white adipose tissue in the testes and epididymis [59,60]. Thus, these tissues were explored to determine adipocyte area (epididymis) and lipid content. For the latter, after 40 days of daily administration of SBHX, the mice were fasted for 5 h and then euthanized. An abdominal incision was made to extract the liver, epididymis, and testes. Each tissue was weighed and kept at −20 °C until the histological process.

#### 4.4.3. Histological and Morphometric Analyses

Testes and epididymal white adipose tissue (eWAT) were dissected, dehydrated, embedded in paraffin, and fixed in formalin for hematoxylin-and-eosin staining [36]. Stained samples of eWAT and testes were evaluated at 40× magnification. For each tissue from each animal, 5 pictures were analyzed. Results were expressed as adipocyte area (µm^2^) and seminiferous tubule cross-sectional area (×1000 µm^2^), as well as seminiferous tubules per field. All procedures were evaluated using the AxioVision software. All histological measurements were performed by two independent observers without knowledge of the source of the tissues, and the results were averaged.

#### 4.4.4. Tissue Lipids

Testes, eWAT, and liver lipids were extracted by the modified Folch technique to determine the percentage of total lipids as previously reported [37]. Individual cholesterol (CHOL) and triglycerides (TG) concentrations were evaluated using an available enzymatic colorimetric kit (DiaSys Diagnostic Systems International, Holsheim, Germany) [40].

### 4.5. Statistical Analysis

Data are expressed as the mean ± standard error of the mean (SEM). One-way ANOVA analysis was performed followed by Dunnett’s post-hoc test or Student’s *t* test. Statistical analysis was performed using SigmaPlot, Systat Software Inc, UK version 11.0, and SigmaStat, version 3.5. *p* < 0.05 was considered significant.

## 5. Conclusions

The neuropharmacological profile of the *B. excelsa* seeds’ nonpolar extract revealed that it produces tranquilizing effects and overweight reduction because of the presence of several fatty acids, evidencing the properties attributed to the *B. excelsa* seeds in folk medicine. No toxic effects per se were observed at any doses evaluated. A GABAergic neurotransmission influence was observed in the presence of *B. excelsa*, as the hypnotic effect of pentobarbital was enhanced. Therefore, it is necessary to carry out further studies on its efficacy and safety in female mice, as well as on the precise mechanism of action involved in the activity of this extract and/or its bioactive components. It is also important to establish regulations for its use in pharmaceutical application.

## Figures and Tables

**Figure 1 molecules-26-03212-f001:**
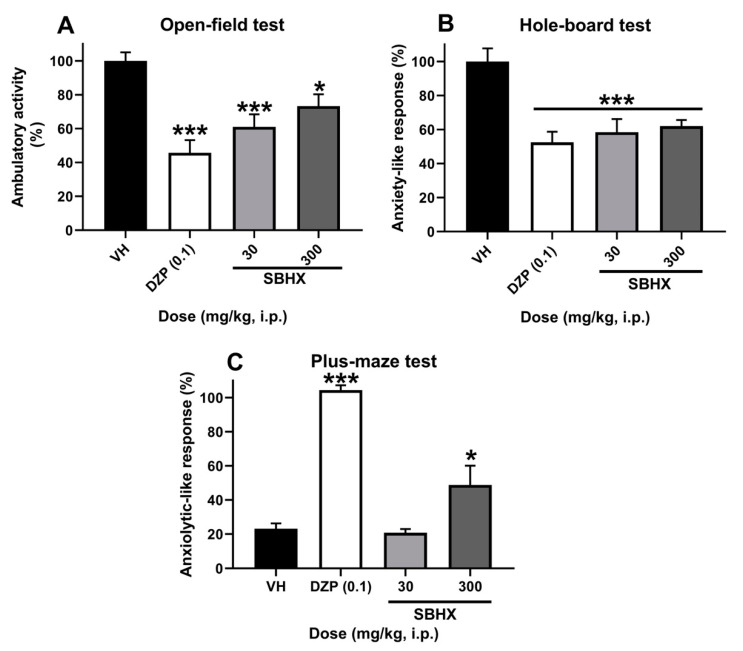
Pharmacological evaluation of the behavioral responses in the open-field (**A**), hole-board (**B**), and elevated plus-maze (**C**) tests in mice receiving SBHX (30 or 300 mg/kg, i.p.) in comparison to the vehicle group and the reference drug diazepam (DZP, 0.1 mg/kg, i.p.). Bars represent the mean ± SEM of five mice (*n* = 5). * *p* <  0.05, *** *p* < 0.001, ANOVA followed by Dunnett’s test. SBHX: Hexane Extract of *B. excelsa* seeds.

**Figure 2 molecules-26-03212-f002:**
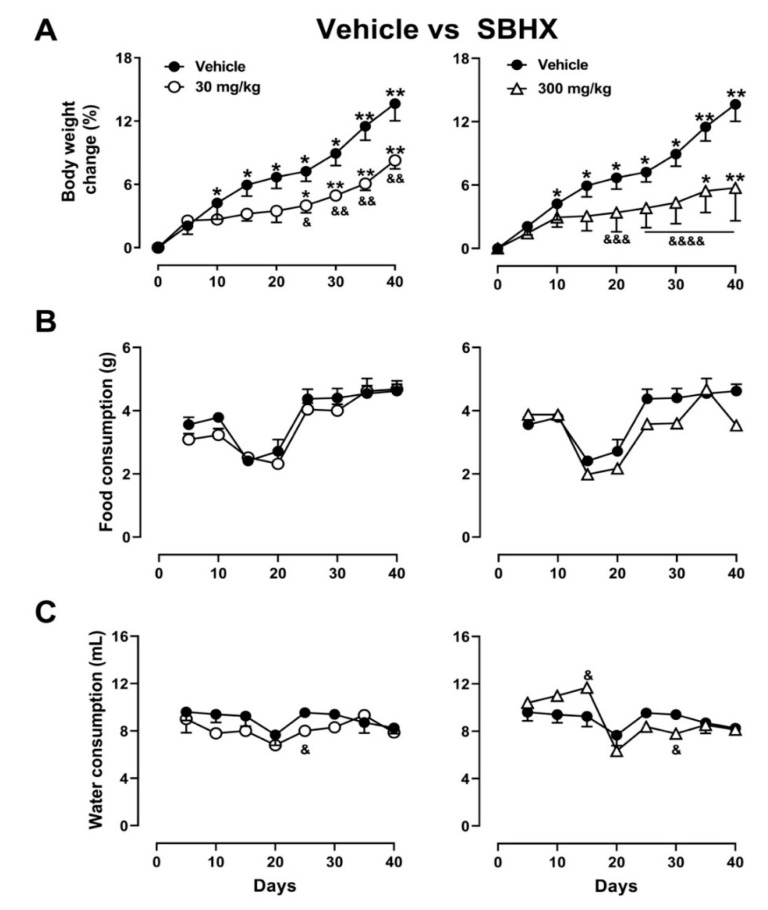
Body weight (**A**), food (**B**), and water consumption (**C**) in mice receiving doses of SBHX (30 and 300 mg/kg, i.p.) in comparison to the vehicle group (VH). Data represent the mean ± SEM of five mice (*n* = 5). ANOVA followed by Dunnett’s test * *p* < 0.05, ** *p* < 0.01 vs. days, or ^&^
*p* < 0.05, ^&&^
*p* < 0.01, ^&&&^
*p* < 0.001, ^&&&&^
*p* < 0.0001 vs. treatment.

**Figure 3 molecules-26-03212-f003:**
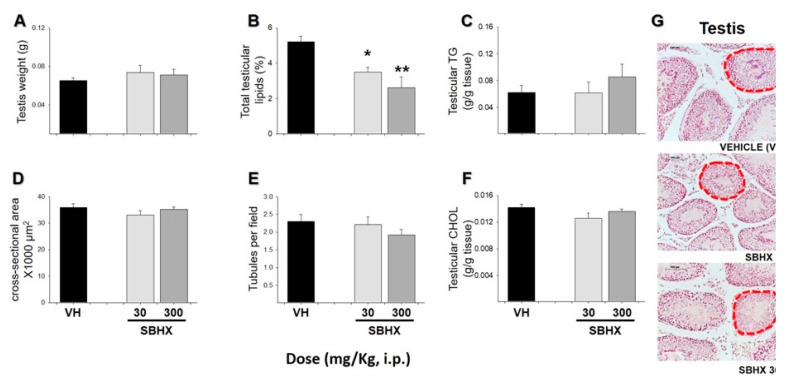
Histological and biochemical analysis of mice testicular tissue receiving SBHX (30 or 300 mg/kg, i.p.) and compared to the vehicle group (VH). Testis weight (g) (**A**), total testicular lipids (%, determined by the Folch method) (**B**), testicular triglycerides (TG, g/g testis) (**C**), seminiferous tubule cross-sectional area (×1000 μm^2^) (**D**), number of seminiferous tubules per field (25 fields observed per treatment) (**E**), testicular cholesterol (CHOL) (**F**). Representative microphotographs of seminiferous tubules taken with a light-field microscope with 40× magnification (**G**). Data represent the mean ± SEM of five mice (*n* = 5). * *p* <  0.05 or ** *p* < 0.01, ANOVA followed by Dunnett’s test.

**Figure 4 molecules-26-03212-f004:**
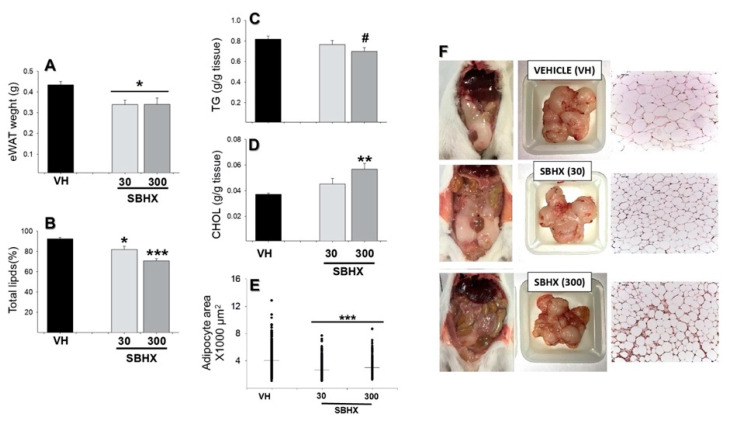
Histological and biochemical analysis of epididymal white adipose tissue (eWAT) of mice receiving SBHX (30 or 300 mg/kg, i.p.) and compared to the vehicle group (VH). eWAT weight (g) (**A**), total epididymal lipids (%, determined by the Folch method) (**B**), epididymal triglycerides (g TG/g tissue) (**C**), epididymal cholesterol (g CHOL/g tissue) (**D**), and adipocyte area (×1000 μm^2^) (**E**). Representative images of abdominal fat, epididymal adipose tissue after being removed, and adipocytes taken with a light-field microscope with 40× magnification (**F**). Data represent the mean±SEM of five mice (*n* = 5) or the median in panel (**E)**. * *p* < 0.05, ** *p* < 0.01 or *** *p* < 0.001, ANOVA followed by Dunnett’s test. # *p* < 0.05, Student’s *t* test.

**Figure 5 molecules-26-03212-f005:**
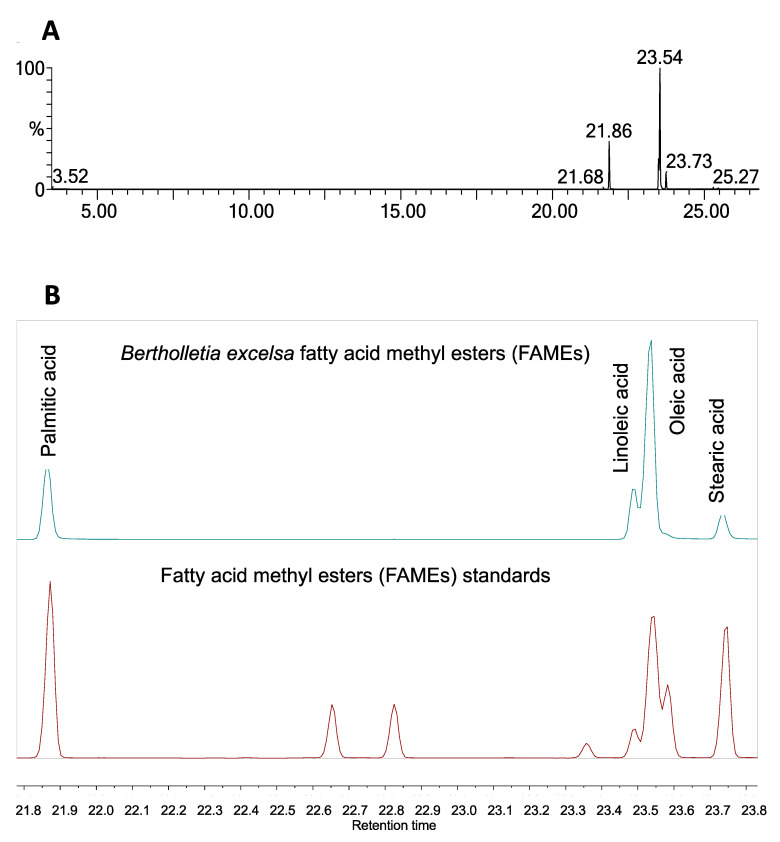
GC-MS chromatogram of B. excelsa oil (**A**) and major fatty acid methyl esters identified and compared to the standards (**B**).

**Figure 6 molecules-26-03212-f006:**
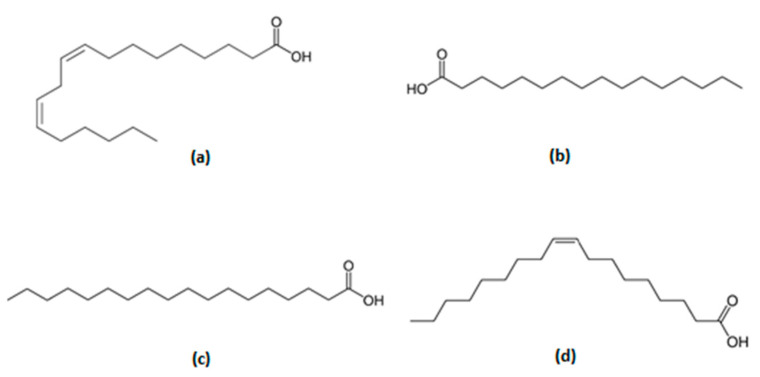
Chemical structure of (**a**) linoleic acid, (**b**) palmitic acid, (**c**) stearic acid, and (**d**) oleic acid as main constituents of the *B. excelsa* oil.

**Table 1 molecules-26-03212-t001:** Sodium pentobarbital (42 mg/kg)-induced hypnotic potentiation in the presence of the *Bertholletia excelsa* seed extract (SBHX) and diazepam (DZP) in mice.

Treatment	Dose (mg/kg)	Latency (min)		Duration of the Hypnosis (min)
		Sedation	Hypnosis	
SP + VH	-	1.22 ± 0.04	3.19 ± 0.21	17.92 ± 04.19
SP + SBHX	0.1	1.10 ± 0.08 *	1.98 ± 0.12	40.80 ± 03.27 *
SP + SBHX	3	1.23 ± 0.14	3.32 ± 0.25	64.32 ± 06.47 *
SP + SBHX	10	1.05 ± 0.09	2.65 ± 0.22	74.04 ± 17.91 *

* *p* < 0.05, ANOVA followed by Dunnett’s test. Treatments (*n* = 6) and vehicle (VH, *n* = 9).

## Data Availability

The data presented in this study are available on request from the corresponding author. The data are not publicly available due to are part of other projects.

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
