# Peer review of "Bertholletia excelsa Seeds Reduce Anxiety-Like Behavior, Lipids, and Overweight in Mice"

_molecules, 2021, doi:10.3390/molecules26113212_

Round 1

Reviewer 1 Report

The manuscript submitted by the authors presents relevant data for a future contribution to the herbal medicines development. However, the manuscript can be improved with some adjusts.

               The authors claim in this work the “anxiolytic, sedative, analgesic and anticonvulsant, as well as changes of the body weight and lipid metabolism in experimental model 77 of obesity induced by diet”.  Do the tests performed really allow this statement? Or do they just indicate these potentiality?

               How were doses of 30 and 300 mg / kg assigned? Is there any literature indicating an ideal range?               Authors should mention the meaning of eWAT

               The authors could better discuss the reason for the choice of the tests directed to the males.

               It is not clear how the authors obtain the hexane fraction yield.The ideal would be to mention how much starting material they had, and how many times the process was repeated, for how long the maceration extraction was maintained. How they eliminate the solvent.

               The mass spectrometry detector is not ideal for estimating the percentage of compounds present in the hexane fraction. They must do it by FID.               The authors must include the total chromatogram of the analysis by GC-MS. And not only on the part of FAME.               It is possible to identify several other compounds present there through the NIST library and as well as hydrocarbon injection. It is not necessary to include the pattern chromatogram. Only indicate on the chromatogram which compounds you can identify.

Author Response

Reviewer #1.

Comments and Suggestions for Authors

The manuscript submitted by the authors presents relevant data for a future contribution to the herbal medicines development. However, the manuscript can be improved with some adjusts.

               The authors claim in this work the “anxiolytic, sedative, analgesic and anticonvulsant, as well as changes of the body weight and lipid metabolism in experimental model of obesity induced by diet”.  Do the tests performed really allow this statement? Or do they just indicate these potentiality?

ANSWER.- These are potential properties of B. excelsa seeds because it produced a depressant activity on the central nervous system. Given that this investigation it was essentially demonstrated the sedative and anxiolytic like-responses of B. excelsa, these activities were emphasized in the text and the words analgesic and anticonvulsant were removed to avoid confusion (line 34).

               How were doses of 30 and 300 mg / kg assigned? Is there any literature indicating an ideal range?              

ANSWER.- In our experience by investigating natural products, we have explored and observed effects in a total range of 3 to 1000 mg/kg depending on the studied plant species and the part of the plant. In the present experimental design, we chosen intermediate doses from this total range to explore neuropharmacological and metabolic responses of B. excelsa. Two doses were considered in order to corroborate not only significant effects but also to know if these effects are produced in a dose-dependent manner (text included in lines 202-204). Two lower doses were used to explore potentiation of the sedative effects when combined with an effective barbituric with sedative properties.

 Authors should mention the meaning of eWAT

ANSWER.- The meaning of eWAT (epididymal white adipose tissue) was included the first time it was mentioned in the text (line 129).

               The authors could better discuss the reason for the choice of the tests directed to the males.

ANSWER.- In this preliminary study with B. excelsa, we have included male mice not only to reduce biological variability because of the close relationship between hormone regulation and lipid metabolism, but also to take advantage that in overweight and obesity exist accumulation of white adipose tissue in the testis and epididymis (lines 237-240, references 36 and 37). Nevertheless, after being obtained a positive anxiolytic-like response, it will be interesting to study the effects of this treatment in female mice in the future in the fact that the prevalence rate of anxiety disorders is approximately twice as high as in men. This information was discussed in the text (lines 241-243, reference 38).

               It is not clear how the authors obtain the hexane fraction yield. The ideal would be to mention how much starting material they had, and how many times the process was repeated, for how long the maceration extraction was maintained. How they eliminate the solvent.

ANSWER.- The extraction process was emphasized in the text as follows: B. excelsa seeds were cut into very small pieces and macerated in hexane (25 mL) during 72 h each time and for three simultaneous extractions. After total evaporation of the solvent using rotatory evaporation and aeration, a translucent oil was obtained with a yield of 55 ± 2.5% (lines 359-363).

               The mass spectrometry detector is not ideal for estimating the percentage of compounds present in the hexane fraction. They must do it by FID. The authors must include the total chromatogram of the analysis by GC-MS. And not only on the part of FAME. It is possible to identify several other compounds present there through the NIST library and as well as hydrocarbon injection. It is not necessary to include the pattern chromatogram. Only indicate on the chromatogram which compounds you can identify.

ANSWER.-  Our apologize but we have no access to an FID equipment at the moment to compare the results found in the present study using the GC-MS technique. However, it is known that GC-MS allows analyze fatty acids either as free fatty acids or as fatty acid methyl esters. A comparison of these two techniques have been already reported to describe certain advantages of GC-MS against FID by concluding that the sensitivity and selectivity of GC–MS make it an advantageous platform for FAME quantification (Dodds et al., 2005), where GC–MS offered two powerful advantages over FID such as the ability to confirm the identity of analytes based on spectral information in addition to retention time, and the ability to separate peaks from a noisy background or coeluting peaks if unique ions are available. These characteristics, taken together with good quantitative performance and the widespread availability of GC–MS instruments offer compelling motivation to predict that GC–MS will eventually become a far more widely exploited alternative to GC-FID for FAME analysis, both quantitative and qualitative.

A justification of this technique was included in the text of results (lines 177-183) supported by the reference [24] (Dodds, E.D.; McCoy, M.R.; Rea, L.D.; Kennish, J.M. Gas chromatographic quantification of fatty acid methyl esters: Flame ionization detection vs electron impact mass spectrometry. Lipids 2005, 40, 419-428. https://doi.org/10.1007/s11745-006-1399-8).

Reviewer 2 Report

The present study evaluated the neuropharmacological profile of an extract of B. excelsa seeds to give evidence of its anxiolytic effect, as well as changes of the body weight and lipid metabolism in an experimental model of obesity-induced by diet. The article reports the results of a well-planned research study, however, some points have to be addressed:

- There is no information regarding the age of the animals.

- Authors should provide the composition of the high-fat diet. Were the experimental diets manufactured following the AIN-93M? It should be provided in the methods.

 - There is no information in the abstract when the animals started to receive the HF diet and for how long? In the methods, the authors describe 40 days of treatment with SBHX. Therefore how do the authors explain the following phrase in the discussion section (lines 264-266)?: “Our results showed after a gross analysis of seminiferous tubules, no changes in the number, this lack of effect could be related to the chronicity of treatment administration, in the present study were 14 days”

- Extract preparation: Please provide the origin of the B. Excelsa Bonpl. seeds. For the production of SBHX, did the authors use the same batch of seeds during the whole experimental period? Was all SBHX extracted at once?

- Please clearly state the number of animals per group. There is no indication regarding the numbers of animals in each group (n=?). There is no information in the methods, results, and legends of the figures. I am also not sure how the authors assigned all animals to each experiment.

- Only male offspring were examined, which is a significant limitation of the study. The females are at increased risk for developing stress-associated pathology such as anxiety and depression with a female/male risk ratio of ~2:1 (Reynolds et al., World Psychiatry. 2015; 14: 74–81).

-Information on light intensity in the open-field apparatus is missing. Also, the time of the day when behavioral testing was done is not mentioned.

 -I suggest counting and add into the Results also the time spent in the peripheral and central area of the open field.

- How do the authors explain the increase in cholesterol levels of epididymal WAT of mice receiving 300 mg/Kg of SBHX and at the same time the total lipids are decreased with this dosage? This point is not discussed.

-In figure 3A replace “Testes” with “Testis 

Author Response

Reviewer #2

The present study evaluated the neuropharmacological profile of an extract of B. excelsa seeds to give evidence of its anxiolytic effect, as well as changes of the body weight and lipid metabolism in an experimental model of obesity-induced by diet. The article reports the results of a well-planned research study, however, some points have to be addressed:

- There is no information regarding the age of the animals.

ANSWER.- Male mice were used from an age of six weeks old at the beginning of the study (line 342).

- Authors should provide the composition of the high-fat diet. Were the experimental diets manufactured following the AIN-93M? It should be provided in the methods.

ANSWER.- Mice were fed with standard diet Purina 5001 (LabDiet Rodent) which calories are provided by protein (28.67%), fat (ether extract, 13.38%), and carbohydrates (57.94%). Water containing sucrose at 34% m/v was included to generate an hypercaloric diet, a combination that induce overweight and obesity in mice (lines 407-409, reference 40).

 - There is no information in the abstract when the animals started to receive the HF diet and for how long? In the methods, the authors describe 40 days of treatment with SBHX. Therefore how do the authors explain the following phrase in the discussion section (lines 264-266)?: “Our results showed after a gross analysis of seminiferous tubules, no changes in the number, this lack of effect could be related to the chronicity of treatment administration, in the present study were 14 days

ANSWER.- In the abstract section, it was emphasized for how long mice received chronic administration of SBHX. The mistake “14 days” was corrected for “40 days” and the text was rewritten to be clearer (lines 299-300).

- Extract preparation: Please provide the origin of the B. Excelsa Bonpl. seeds. For the production of SBHX, did the authors use the same batch of seeds during the whole experimental period? Was all SBHX extracted at once?

ANSWER.- The extraction process was emphasized in the text as follows:

SBHX was obtained from commercial presentations of B. excelsa seeds that are bought by people from SdB Guadalajara, Mexico, and taken as a nutritional supplement.

Each presentation contains approximately 1.5 g of B. excelsa seeds. To prepare SBHX we used several presentations to know the average yield of the extract, for this B. excelsa seeds were cut into very small pieces and macerated in hexane (25 mL) during 72 h each time and for three simultaneous extractions for each sample. After total evaporation of the solvent using rotatory evaporation and aeration, a translucent oil was obtained with a yield of 55 ± 2.5% (lines 359-363). Then, all the SBHX extracted was used for the biological experiments.

- Please clearly state the number of animals per group. There is no indication regarding the numbers of animals in each group (n=?). There is no information in the methods, results, and legends of the figures. I am also not sure how the authors assigned all animals to each experiment.

ANSWER.- Number of animals per group were indicated in point 4.1 animals: groups of at least 5 animals (n>5).  (line 344) and in the legends of the figures (lines 101, 126, 152, 161). It was also included in the table 1 (line 121).

- Only male offspring were examined, which is a significant limitation of the study. The females are at increased risk for developing stress-associated pathology such as anxiety and depression with a female/male risk ratio of ~2:1 (Reynolds et al., World Psychiatry. 2015; 14: 74–81).

ANSWER.- We agree to your comment. .- In this preliminary study with B. excelsa, we have included male mice not only to reduce biological variability by sex because of the close relationship between hormone regulation and lipid metabolism, but also to take advantage that in overweight and obesity exist accumulation of white adipose tissue in the testis and epididymis (lines 237-240, references 36 and 37). Nevertheless, after being obtained a positive anxiolytic-like response, it will be interesting to study the effects of this treatment in female mice in the future in the fact that the prevalence rate of anxiety disorders is approximately twice as high as in men. This information was discussed in the text (lines 241-243, reference 38).

-Information on light intensity in the open-field apparatus is missing. Also, the time of the day when behavioral testing was done is not mentioned.

ANSWER.- Information was included as follows:
Experiments were done in a schedule of 9:00 to 14:00 h (line 381). Anxiety-like behavior was observed using dim red light of
30 lux (line 384).

 -I suggest counting and add into the Results also the time spent in the peripheral and central area of the open field.

ANSWER.- The open-field test was applied as adjuvant assay of the hole-board and elevated plus-maze tests to corroborate ambulatory activity of mice. Because of this, it was reported in percentage of the total count of explored squares. But in agreement to your comment, it should be registered the peripheral and central activity mainly to take advantage of data not only for anxiolytic or sedative-like response but also to suggest a possible mechanism of action in future experiments.

- How do the authors explain the increase in cholesterol levels of epididymal WAT of mice receiving 300 mg/Kg of SBHX and at the same time the total lipids are decreased with this dosage? This point is not discussed.

ANSWER: We have included an argument in discussion section as follows:

Total percentage of lipids was significatively reduced in the presence of B. excelsa, in part because of a reduction on TGs, which represent the main lipid component of dietary fat and fat depots of animals (included in lines 260-262). On the other hand, an increase in CHOL was observed in mice receiving chronic administration at the highest dosage (300 mg/kg) (lines 289-290). Given that CHOL is an unsaturated alcohol of the steroid compounds, essential for cell membranes, but also a precursor of adrenal and gonadal steroid hormones, it suggests its involvement in reproductive activity (lines 291-293). It is common that some components on the lipid fraction increase or decrease depending on different factors such as a relationship with lipid diet content, metabolic ageing, and sex (lines 287-289, references 36,37,46). In this study we explored CHOL and TGs from the total percentage of lipids because they are two important and major constituents of the lipid fraction in the body (lines 253-254, reference 41). However, it would be interesting to explore other components of the total lipid fraction in future studies since according to the present results we cannot discard that some of them are likely implicated in the effects of B. excelsa (lines 255-257).

ANSWER.- This mistake was corrected in figure 3A (line 143).

Round 2

Reviewer 1 Report

The authors must include the total chromatogram of the analysis by GC-MS. And not only on the part of FAME. It is possible to identify several other compounds present there through the NIST library and as well as hydrocarbon injection. It is not necessary to include the pattern chromatogram. Only indicate on the chromatogram which compounds you can identify.

Author Response

The authors must include the total chromatogram of the analysis by GC-MS. And not only on the part of FAME. It is possible to identify several other compounds present there through the NIST library and as well as hydrocarbon injection. It is not necessary to include the pattern chromatogram. Only indicate on the chromatogram which compounds you can identify.

R=Thank you for your suggestion and comment, we are including the GC-MS chromatogram where the most abundant fatty acids were identified. Our results agreed with the phytochemical studies previously reported in the literature by Peña-Muñiz et al., 2015 (Reference 23).

Reviewer 2 Report

The authors have addressed the initial critiques adequately. All of my concerns were well addressed. I do not have any other comments.

Author Response

The authors have addressed the initial critiques adequately. All of my concerns were well addressed. I do not have any other comments.

R= thanks for your comments

This manuscript is a resubmission of an earlier submission. The following is a list of the peer review reports and author responses from that submission.